# Recombination Flow Matching Model for Protein Evolution

## Abstract

The design of novel proteins, distinct from those found in nature, holds immense potential for advancing drug discovery, biotechnology, and material science. However, current methodologies often face significant limitations in generating both novel protein structures. Biological evolution, a natural process that fosters novelty, heavily relies on recombination—yet this mechanism remains largely untapped in protein design. In this work, we propose the Recombination Flow Matching (RFM) model, a novel generative model inspired by the principles of evolution. RFM meticulously preserves the structural integrity of selected recombination protein segments during recombination while autonomously optimizing their spatial arrangement within the resultant protein. Using a common benchmark dataset, we demonstrate that RFM significantly outperforms established methods in producing structurally novel proteins. This approach opens new frontiers in protein design, leveraging evolutionary recombination to enhance the novelty of protein design. To the best of our knowledge, RFM is the first model to incorporate recombination into protein design.

## 1 Introduction

Proteins serve as the molecular engines of life, performing a diverse array of functions (Correia et al., 2014; Linsky et al., 2020; Sesterhenn et al., 2020; Jiang et al., 2008). Exploring the vast array of possible protein structures is vital for creating new therapeutics and materials. This has led to a surge of interest in designing synthetic proteins that differ from those found in nature (Callaway, 2022; Watson et al., 2023; Ingraham et al., 2023; Madani et al., 2023). The creation of novel genes, proteins, and metabolic pathways is often driven by evolutionary processes (Levy, 2019; Chen et al., 2013), which foster both novelty and diversity. Recombination is one of the key mechanisms in biological evolution (Barton & Charlesworth, 1998; Otto & Lenormand, 2002; Netzer & Hartl, 1997). However, its application in protein design remains largely untapped (Wang et al., 2024; Nordwald et al., 2013). Existing methods often struggle to generate truly novel protein structures as they often fail to fully take the natural biological evolution process into consideration.

Previous research has explored generative models for protein design, with a particular focus on techniques such as flow matching (Bose et al., 2024; Yim et al., 2024; Jing et al., 2024) and diffusion models (Ingraham et al., 2023; Watson et al., 2023; Yim et al., 2023). However, these models tend to learn the distribution of protein structures by concentrating solely on individual residues, overlooking the significance of larger protein segments. Inpainting-based methods (Watson et al., 2023; Ingraham et al., 2023) and conditional generation approaches (Trippe et al., 2023; Yim et al., 2024) attempt to address this limitation by incorporating protein segments into their models. Despite these efforts, inpainting-based techniques require the manual identification of segment positions within the resulting protein structure, which is not readily available. Moreover, recombination often involves multiple segments, requiring optimization of their relative positions, which is beyond the capabilities of inpainting methods. Conditional generation methods, meanwhile, face challenges in preserving the structural integrity of the segments to be recombined.

To address the challenges of generating novel proteins, we propose a novel model dubbed **R**ecombination **F**low **M**atching (**RFM**), which recombines protein segments to generate innovative protein structures. RFM is inspired by the principles of biological evolution, particularly recombination, and is designed to ensure the structural integrity of the segments being recombined. Moreover,

RFM automates the optimization of segment positions within the resultant protein structure. We evaluate the performance of RFM on benchmark datasets of protein structures and demonstrate that it surpasses existing methods in its ability to generate novel proteins.

RFM is designed to address the limitations of existing methods by incorporating the following key features: First, RFM leverages principles from biological evolution, particularly recombination, to generate novel proteins. Second, to ensure structural integrity, RFM treats the protein segments intended for recombination as rigid bodies. Lastly, RFM automates the optimization of segment positions within the resultant protein structure by adhering to rigid body dynamics, ensuring the integrity of the segments. To demonstrate the effectiveness of RFM, we conduct experiments on benchmark datasets of protein structures. Extensive experiments show RFM is capable of recombining proteins to generate novel structures, outperforming existing methods. Besides, trained on recombining two proteins, RFM can generalize to recombine multiple proteins, which indicates RFM learned the underlying principles of recombination.

To the best of our knowledge, RFM is the first model to incorporate recombination into protein design for enhancing the novelty of protein structures. The main contributions of our work can be summarized as follows:

- We propose Recombination Flow Matching (RFM) model, a novel generative model for protein design that leverages recombination to generate novel protein structures.

- RFM preserves the structural integrity of protein segments intended for recombination and automates the process of designing their optimal positions within the resultant protein structure.

- We demonstrate the effectiveness of RFM on benchmark datasets of protein structures, showing that it outperforms existing methods in generating novel proteins.

## 2 RELATED WORKS

### 2.1 GENERATIVE MODELS FOR PROTEIN DESIGN

Deep generative models have been extensively explored for protein design, with approaches such as flow matching models (Bose et al., 2024; Yim et al., 2024; Jing et al., 2024) and diffusion models (Ingraham et al., 2023; Watson et al., 2023; Yim et al., 2023; Liu et al., 2024; Trippe et al., 2023). These models learn the distribution of protein structures based solely on individual residues. However, they tend to overlook the significance of larger protein segments, which are critical for protein functions. As a result, these models frequently struggle to generate truly novel protein structures. In response, our RFM model addresses this limitation by incorporating protein segments, drawing inspiration from recombination—a key process in biological evolution.

### 2.2 PROTEIN RECOMBINATION

Recombination is a key mechanism driving biological evolution, yet it remains underexplored in protein design (Netzer & Hartl, 1997). Previous works, including inpainting and conditional generation methods, have attempted to incorporate protein segments into generative models for protein recombination. Despite this, Inpainting-based methods (Watson et al., 2023; Ingraham et al., 2023) require manual identification of segment positions within the resulting protein structure, which is not readily available. Moreover, recombination often involves multiple segments, requiring optimization of their relative positions, which is beyond the capabilities of inpainting methods. Conditional generation methods (Trippe et al., 2023; Yim et al., 2024) yet face challenges in preserving the structural integrity of the segments to be recombined. Our RFM preserves the structural integrity of protein segments while automating the process of designing their optimal positions within the resultant protein structure.

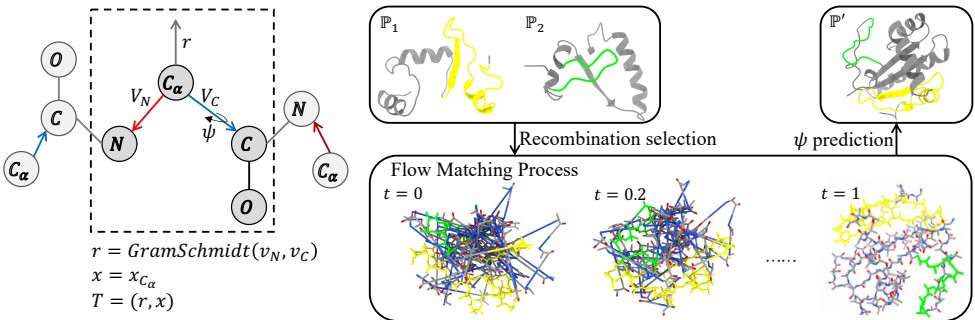

Figure 1: Overview of the residue representation and Recombination Flow Matching (RFM) model. Left: Protein backbone representation. A residue is represented with a rotation $r$ and a translation $x$. Rotation $r$ is parameterized as a matrix $\boldsymbol{R}$ and translation is parameterized as a vector $\boldsymbol{x}$, where the rotation is the position of $\alpha$ carbon. Right: RFM model architecture. The process includes recombination selection, flow matching, and $\psi$ prediction.

## 3 PRELIMINARIES

### 3.1 PROTEIN RECOMBINATION

We define a protein $\mathbb{P} = \{\boldsymbol{A}, \boldsymbol{T}\}$ as a sequence of amino acids (residues) $\boldsymbol{A} = [a_1, a_2, \ldots, a_n] \in \mathbb{R}^n$ and their 3D positions $\boldsymbol{T} = [\boldsymbol{T}_1, \boldsymbol{T}_2, \ldots, \boldsymbol{T}_n] \in \mathbb{R}^{n \times 4 \times 4}$, where $n$ denotes the number of residues in the protein. $a_i \in \mathbb{A}^{20}$ denotes the type of residue. $\boldsymbol{T}_i \in \mathbb{R}^{4 \times 4}$ is the pose matrix of the residue which can be represented as a rotation matrix $\boldsymbol{R}_i \in \mathbb{R}^{3 \times 3}$ and a translation $\boldsymbol{x}_i \in \mathbb{R}^3$. The pose matrix $\boldsymbol{T}_i$ can be decomposed as $\boldsymbol{T}_i = [\boldsymbol{R}_i, \boldsymbol{x}_i]$. In this work, we focus on the protein backbone generation $\boldsymbol{T}$, where the residue types are ignored and the protein structure is represented by the backbone atoms, following previous works (Watson et al., 2023; Yim et al., 2023).

Protein recombination involves the exchange of segments between two or more proteins to generate novel protein structures. Given $m$ proteins $[\mathbb{P}_1, \mathbb{P}_2, \ldots, \mathbb{P}_m]$, we aim to recombine them to generate a novel protein $\mathbb{P}'$. The recombination process involves selecting segments from $[\mathbb{P}_1, \mathbb{P}_2, \ldots, \mathbb{P}_m]$ and combining them to form $\mathbb{P}'$. We define the protein recombination as follows:

**Definition 3.1** (Protein Recombination). Given $m$ proteins $[\mathbb{P}_1, \mathbb{P}_2, \ldots, \mathbb{P}_m]$, the protein recombination process aims to generate a novel protein $\mathbb{P}'$ by selecting segments from $[\mathbb{P}_1, \mathbb{P}_2, \ldots, \mathbb{P}_m]$ and recombining them, *i.e.*, $\mathbb{P}' = f(\mathbb{P}_1, \mathbb{P}_2, \ldots, \mathbb{P}_m)$, where $f$ is the recombination function.

### 3.2 PROTEIN BACKBONE PRAMETERIZATION

The protein backbone parameterization follows previous works (Yim et al., 2023; 2024). Each residue in the backbone is parameterized as an orientation preserving rigid transformation (*Frame*) $\boldsymbol{T} = (\boldsymbol{R}, \boldsymbol{x})$ as shown in Fig. 1 (left). The rotation matrix $\boldsymbol{R} \in \mathrm{SO}(3)$ represents the rotation $r$ which is obtained through the Gram-Schmidt process. The translation $\boldsymbol{x} \in \mathbb{R}^3$ is the position of $\alpha$ carbon in 3D space. Therefore, the atom coordinates of the $i$-th residue in the 3D space can be obtained through the transformation $\boldsymbol{T}_i$ as follows:

$$[\mathrm{N}_i, \mathrm{C}_i, (\mathrm{C}_\alpha)_i] = \boldsymbol{T}_i[\mathrm{N}^*, \mathrm{C}^*, (\mathrm{C}_\alpha)^*], \tag{1}$$

where $\mathrm{N}^*, \mathrm{C}^*, (\mathrm{C}_\alpha)^* \in \mathbb{R}^3$ are the fixed coordinates centered at $(\mathrm{C}_\alpha)^* = (0, 0, 0)$. The position of the atom Oxygen is determined as the torsion angle of the bond between $\alpha$-carbon and the carbon is given. Since the transformation $\boldsymbol{T}$ can be decomposed as a rotation $\boldsymbol{R}$ and a translation $\boldsymbol{x}$. We can rewrite the transformation separately as follows:

$$[\mathrm{N}_i, \mathrm{C}_i, (\mathrm{C}_\alpha)_i] = \boldsymbol{R}_i[\mathrm{N}^*, \mathrm{C}^*, (\mathrm{C}_\alpha)^*] + \boldsymbol{x}_i, \tag{2}$$

Therefore the flow matching model can perform on the two manifolds of rotation and translation.

### 3.3 Flow Matching for Protein Design

Flow matching is a generative model that learns the distribution of protein structures by matching the flow of the protein backbone (Bose et al., 2024; Yim et al., 2024). Given a prior distribution $P$, the flow matching model learns a flow $\Phi$ that maps the distribution to the distribution $P_T$ of protein backbone $T$ as shown in Fig. 1(right) (Yim et al., 2024; Chen & Lipman, 2023). Given a conditional flow $T_t = \Phi_t(T_0|T_1)$, which is along the geodesic path between $T_0$ and $T_1$:

$$T_t = \exp_{T_0}(t \log_{T_0} T_1). \tag{3}$$

For the manifolds of rotation and translation, the flow is simplified as:

$$\boldsymbol{R}_t = \exp_{\boldsymbol{R}_0}(t \log_{\boldsymbol{R}_0} \boldsymbol{R}_1), \tag{4}$$

$$\boldsymbol{x}_t = (1-t)\boldsymbol{x}_0 + t\boldsymbol{x}_1. \tag{5}$$

Finally, the objective of the flow matching model is to match the flow of the protein backbone, which is defined as:

$$\mathcal{L}_{fm} = \mathbb{E}_{t, P_1(T_1), P_0(T_0)} \left[ \sum_{n=1}^{N} \left\{ \left\| v_{\boldsymbol{x}}^{(n)}(T_t, t) - \dot{\boldsymbol{x}}_t^{(n)} \right\|_{\mathbb{R}^3}^2 + \left\| v_{\boldsymbol{R}}^{(n)}(T_t, t) - \dot{\boldsymbol{R}}_t^{(n)} \right\|_{SO(3)}^2 \right\} \right], \tag{6}$$

where $t \sim \mathcal{U}(0, 1-\epsilon)$ for $\epsilon$ small enough, here we set $\epsilon = 10^{-3}$. $v_{\boldsymbol{x}}^{(n)}$ and $v_{\boldsymbol{R}}^{(n)}$ are the vector field. $n$ denotes the $n$-th residue. Following the definition of $\boldsymbol{x}_t^{(n)}$ and $\boldsymbol{R}_t^{(n)}$, the optimization object is simplified as:

$$\mathcal{L}_{fm} = \mathbb{E}_{t, P_1(T_1), P_0(T_0)} \left[ \frac{1}{(1-t)^2} \sum_{n=1}^{N} \left\{ \left\| \hat{\boldsymbol{x}}_1^{(n)}(T_t, t) - \boldsymbol{x}_1^{(n)} \right\|_{\mathbb{R}^3}^2 \right. \right.$$

$$\left. \left. + \left\| \log_{\boldsymbol{R}_t^{(n)}} \left( \hat{\boldsymbol{R}}_1^{(n)}(T_t, t) \right) - \log_{\boldsymbol{R}_t^{(n)}} \left( \boldsymbol{R}_1^{(n)} \right) \right\|_{SO(3)}^2 \right\} \right], \tag{7}$$

where $\hat{\boldsymbol{x}}_1^{(n)}$ and $\hat{\boldsymbol{R}}_1^{(n)}$ are the predicted translation and rotation of the $n$-th residue respectively.

### 3.4 Euler Integration for Inference

In the inference process, the flow matching model uses Euler integration to generate the protein structure iterative as follows:

$$\boldsymbol{R}^{(t)} = \hat{\boldsymbol{R}}(\boldsymbol{R}^{(t-1)})^{-1}\Delta t/(1-t)\boldsymbol{R}^{(t-1)}$$

$$\boldsymbol{x}^{(t)} = (\hat{\boldsymbol{x}} - \boldsymbol{x}^{(t-1)})\Delta t/(1-t) + \boldsymbol{x}^{(t-1)}, \tag{8}$$

where $\hat{\boldsymbol{R}}$ and $\hat{\boldsymbol{x}}$ are the predicted rotation and translation. $\Delta t$ is the step size of the integration. The $\Delta t/(1-t)$ is the scaling factor that performs on rotation vectors following (Yim et al., 2024).

## 4 Recombination Flow Matching

To explore the vast space of protein structures, we propose Recombination Flow Matching (RFM), a generative model that leverages recombination to generate novel protein structures. RFM is inspired by the principles of biological evolution, particularly recombination, and is designed to ensure the structural integrity of segments being recombined. Moreover, RFM automates the optimization of segment positions within the resultant protein structure. The overview of the RFM model is shown in Fig. 2. The RFM model consists of two main components: recombination selection and rigid dynamics involved flow matching. First of all, the recombination selection component selects segments from multiple proteins $[\mathbb{P}_1, \mathbb{P}_2, \cdots, \mathbb{P}_m]$ to be recombined. The selected segments are then input into the rigid dynamics involved flow matching process, where the flow matching model is used to recombine the segments and generate a novel protein structure.

Figure 2: Overview of the Recombination Flow Matching (RFM) model. The recombination process involves selecting segments from multiple proteins $\mathbb{P}_1$ and $\mathbb{P}_2$ and recombining them to generate a novel protein $\mathbb{P}'$. The RFM model consists of three components: recombination selection and rigid dynamics involved flow matching. The yellow and green colors indicate the selected recombination segments and the gray color indicates the other residues.

### 4.1 RECOMBINATION SELECTION

Given $m$ proteins $[\mathbb{P}_1, \mathbb{P}_2, \cdots, \mathbb{P}_m]$, the recombination selection aims to select segments from these proteins to be recombined. Since we ignore the residue types and focus on the protein backbone, the recombination selection process involves selecting segments based on the protein backbone. We denote the selected segments as $T_S = [T_{s1}, T_{s2}, \cdots, T_{sk}]$, where $k$ is the number of segments to be recombined. $T_{sk} = [\boldsymbol{T}_0, \boldsymbol{T}_1, \cdots, \boldsymbol{T}_n]$ denotes the segment from the $k$-th protein consists of $n$ residues, where $\boldsymbol{T}_i$ is the pose matrix of the $i$-th residue in the segment.

In the resultant protein $\mathbb{P}'$, which can also be denoted as $T$, we denote the selected segments as $\mathbb{T}_S$ and the other residues as $T_r$. For simplicity, we denote all the other residues from the selected segments as $T_s$. In the training process, we randomly select $k$ segments from one single protein as one training sample following (Watson et al., 2023), where the protein is taken as the target protein. In the inference process, we can select multiple segments from different proteins to generate a novel protein following the recombination process.

### 4.2 RIGID DYNAMICS INVOLVED FLOW MATCHING

The rigid dynamics involved flow matching maintains the structural integrity of the segments being recombined while automating the optimization of their positions within the resultant protein structure. Since the translation and rotation can be decomposed as $\boldsymbol{T}_i = [\boldsymbol{R}_i, \boldsymbol{x}_i]$, the flow matching model can be applied to the manifolds of rotation and translation separately. Here, we also introduce the RFM on the manifolds of rotation and translation separately.

#### 4.2.1 CONDITIONAL FLOW

**Rotation manifold.** For rotation, we reform the flow on the segments as follows:

$$R_t = \left[ \overline{R_{s1}^t R_{s0}^{-1}} R_{s0}, \quad R_{r1}^t \right], \tag{9}$$

where the two items indicate the flow for the selected segments and the other residues respectively. $R = [\boldsymbol{R}_1, \boldsymbol{R}_2, \cdots, \boldsymbol{R}_n] \in \mathbb{R}^{n \times 3 \times 3}$ is the tensor of rotation matrices. $R_{s1}$ and $R_{r1}$ denote the rotation tensor of the selected segments and the other residues respectively at time $t = 1$. $R_{s1}^{-1}$ is the inverse of the rotation tensor. $\overline{R_{s1}^{(t-1)}}$ is the average of the rotation tensor of the selected segments at time $t - 1$. The average operation is performed on each selected segment respectively. Generally, we estimate the rigid rotation of the selected segments by considering the rotations of the residues that make up these segments. For efficient computation, we adopt the strategy of averaging the rotation vectors of constituent residues. Without loss of generalization, alternative techniques can also be employed to estimate the rigid rotation of the selected segments.

Specifically, we define a virtual coordinate system for each selected segment with the rotation of an identity $I$ and a translation of the segment geometry center $\boldsymbol{o} = \overline{\boldsymbol{X}_s}$ as its origin. We average the

rotation transformation of selected segments under the virtual coordinate system as Eq. 10. Then the transformation for each residue is obtained by mapping them back to each residue as Eq. 11

$$[\Delta\boldsymbol{R}_s, \Delta\boldsymbol{R}_r] = \left[\overline{\boldsymbol{R}_{s1}^t \boldsymbol{R}_{s0}^{-1}}, \quad \boldsymbol{R}_{r1}^t \boldsymbol{R}_{r0}^{-1}\right] \tag{10}$$

$$\boldsymbol{R}_t = [\Delta\boldsymbol{R}_s, \quad \Delta\boldsymbol{R}_r]\boldsymbol{R}_0 \tag{11}$$

The rotation of a rigid body will cause the translation of its constituent point. Therefore, to maintain the structural integrity of each selected segment, we also calculate the translation of residue accompanying its rotation in the virtual coordinate system following the rigid dynamics as follows:

$$\Delta\boldsymbol{X} = [\Delta\boldsymbol{R}_s(\boldsymbol{X}_s - \boldsymbol{o}) + \boldsymbol{o} - \boldsymbol{X}_s, \quad 0_r] \tag{12}$$

where $\boldsymbol{X} = [\boldsymbol{x}_1, \boldsymbol{x}_2, \cdots, \boldsymbol{x}_n] \in \mathbb{R}^{n\times 3}$ is the translation matrix of the residue.

**Translation manifold.** For translation, the flow is reformed as follows:

$$\boldsymbol{X}_t = \left[\overline{t(\boldsymbol{X}_{s1} - \boldsymbol{X}_{s0}) + \boldsymbol{X}_{s0}}, \quad t(\boldsymbol{X}_{r1} - \boldsymbol{X}_{r0}) + \boldsymbol{X}_{r0}\right] + \Delta\boldsymbol{X} \tag{13}$$

We average the translation of the constituent residues of each selected segment. To maintain the structural integrity of selected segments, we add the translation accompanying the rigid body rotation to the translation transform in the flow.

### 4.2.2 MODEL ARCHITECTURE

Following Yim et al. (2023); Liu et al. (2024); Watson et al. (2023), we update the translation and rotation at each layer. Given the update prediction for rotation $\boldsymbol{R}^{(l-1)}$ and translation $\boldsymbol{X}^{(l-1)}$ as $\hat{\boldsymbol{R}}_\delta$ and $\hat{\boldsymbol{X}}_\delta$, the updated rotation $\boldsymbol{R}^{(l)}$ and translation $\boldsymbol{X}^{(l)}$ can be obtained similar to the process in Eq. 9 and Eq. 13 as follows:

$$\boldsymbol{R}^{(l)} = \left[\left(\overline{\hat{\boldsymbol{R}}_\delta}\right)_s, \quad \left(\hat{\boldsymbol{R}}_\delta\right)_r\right]\boldsymbol{R}^{(l-1)} \tag{14}$$

$$\boldsymbol{X}^{(l)} = \left[\left(\overline{\hat{\boldsymbol{R}}_\delta(\boldsymbol{X}^{(l-1)} - \boldsymbol{o}) + \boldsymbol{o} - \boldsymbol{X}^{(l-1)} + \overline{\boldsymbol{R}^{(l-1)}\hat{\boldsymbol{X}}_\delta}}\right)_s, \quad \left(\boldsymbol{R}^{(l-1)}\hat{\boldsymbol{X}}_\delta\right)_r\right] + \boldsymbol{X}^{(l-1)}. \tag{15}$$

In the update process, we first estimate the rigid rotation of the selected segments by considering the rotations of the residues that make up these segments. Specifically, we apply the rotation update to the frame and calculate the rotation transformation between the original and transformed frames in the virtual coordinate system as $\overline{\hat{\boldsymbol{R}}_\delta}$. The estimated rigid rotation is the average of the rotation transformation of the selected segments, which is further applied to each residue.

Then we calculate the translation of the residue accompanying its rotation in the update process. The translation of the residue is calculated by the rigid dynamics, which is the same as Eq. 12. The predicted translation update is obtained by averaging the translation of the constituent residues of each selected segment as $\overline{\boldsymbol{R}^{(l-1)}\hat{\boldsymbol{X}}_\delta}$. Finally, the new translation consists of the reformed translation update prediction and the translation accompanied by the rotation update.

### 4.2.3 TRAINING LOSS

The training losses consist of two parts, flow matching loss as Eq. 7 and auxiliary structure loss following (Yim et al., 2024). The auxiliary structure loss includes the backbone atom coordinate loss and atom pair distance loss as shown in Appendix. A.2.1. To avoid the structural violations, we introduce additional C-N bond loss to our RFM, which is commonly used in precious works (Jumper et al., 2021; Watson et al., 2023; Liu et al., 2024) as follows:

$$\mathcal{L}_{C-N} = \frac{1}{N_{bonds}} \sum_{i=1}^{N_{bonds}} \max(|l_{pred}^i - l_{lit}^i| - \tau, 0), \tag{16}$$

where $l_{pred}$ is the C-N bond length in the predicted structure, $l_{lit}$ is literature C-N bond length, and $\tau$ is a threshold. $N_{bonds}$ is the number of C-N bonds in the protein structure.

### 4.3 INFERENCE

Euler integration with 100 steps is used in the inference process following previous works (Yim et al., 2024; Chen & Lipman, 2023). Similarly, to maintain the structural integrity of the segments being recombined, we reform the Euler integration step from Eq. 8 as follows:

$$R^{(t)} = \left[ (\Delta R)_s, \quad \left( \hat{R}(R^{(t-1)})^{-1} \Delta t/(1-t) \right)_r \right] R^{(t-1)} \tag{17}$$

$$X^{(t)} = \left[ \left( \Delta R(X^{(t-1)} - o) + o - X^{(t-1)} + \overline{\frac{(\hat{X} - X^{(t-1)})\Delta t}{1-t}} \right)_s, \left( \frac{(\hat{X} - X^{(t-1)})\Delta t}{1-t} \right)_r \right] + X^{(t-1)},$$

where $\Delta R = \overline{\hat{R}(R^{(t-1)})^{-1} \Delta t/(1-t)}$. Besides, we model the problem of determining the position of residues in the protein sequence as a traveling salesman problem (TSP) and solve it using the Concorde solver following (Liu et al., 2024). Specifically, we take the distance from the carbon atom of residue $i$ to the nitrogen atom of residue $j$ as the path between residue $i$ and $j$.

More details of the method can be found in the Appendix. A.1

## 5 EXPERIMENTS

We first pre-train RFM on the task of unconditional protein generation with the protein data bank dataset (Berman et al., 2000) and then fine-tune it on the task of recombination. To demonstrate the effectiveness of RFM, we first evaluate the ability of RFM to recombine different numbers of proteins with the metric of recombination success rate. Then we compare RFM with previous unconditional protein generation methods on novelty metrics.

### 5.1 SETUP

**Dataset.** All the experiments are conducted on the **protein data bank dataset (PDB)** (Berman et al., 2000), which contains 59,1228 proteins with a length of 60 to 512 residues. We construct the training dataset by randomly cropping $m$ segments of length 2 to 40 residues from one single protein to form the selected recombination segments for fine-tuning. Each entry in the training dataset consists of a complete protein structure as the target and $m$ segments as the recombination chosen segments.

**Evaluation metrics.** We mainly employ the **recombination success rate (RSR)** and **Novelty** metrics to evaluate the model performance following previous works (Yim et al., 2023; 2024; Liu et al., 2024). A generation is successful if the produced protein is *designable* and the selected recombination segments are present within the generation. The designability of a generated protein is assessed using the scTM score. **scTM** refers to the TM-score (Zhang & Skolnick, 2005) between the structure of the generated protein and the structure reconstructed through a sequence design model (Protein-MPNN) (Dauparas et al., 2022) and a structure prediction model (Lin et al., 2023). A higher scTM score signifies greater similarity between the two protein structures. A protein is considered designable if scTM $> 0.5$. Since we ensure the existence of selected recombination segments in the generation through the rigid dynamics involved flow matching, the **SRS** is the percentage of designable proteins in the generation. To assess the **Novelty**, we use the pdbTM score, which compares each generated protein structure with those in the Protein Data Bank (PDB) (Berman et al., 2000). The similarity between the two structures is quantified using the TM score, known as the pdbTM score. A generation is classified as a novel if pdbTM $< 0.7$. More details can be found in the Appendix. A.3.1.

**Compared approaches.** We mainly compare our RFM with unconditional generation methods including FrameFlow (Yim et al., 2024), FrameDiff (Yim et al., 2023), and VFNDiff (Mao et al., 2024). FrameFlow is a flow matching model for protein structure generation. FrameDiff and VFN-Diff are two diffusion models for protein generation.

**Training details.** To evaluate the generalization of RFM, RFM is fine-tuned on the proteins with a length less than 256 residues. We train RFM to recombine two, three, four, and five proteins

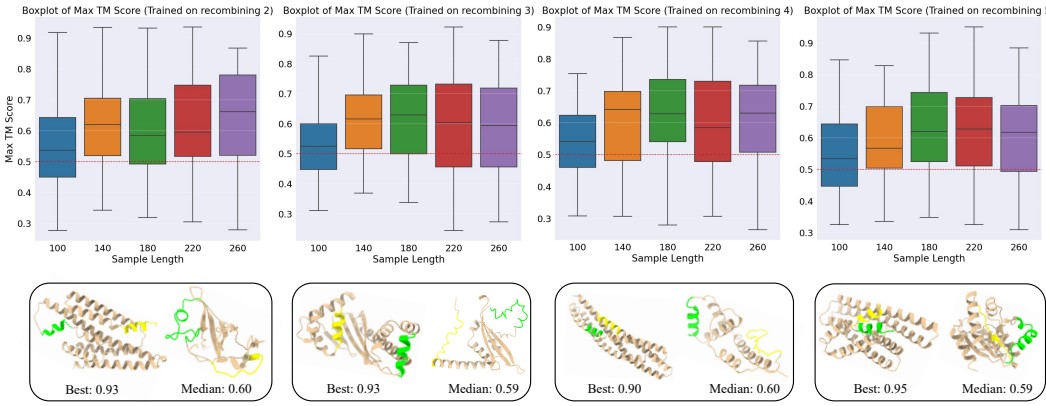

Figure 3: Recombination success rate of RFM trained on recombining different numbers of proteins. The first row shows the boxplot of scTM score of models trained on recombining different numbers of proteins and tested on recombining two proteins. The horizontal and vertical coordinates are the length of the protein and the scTM score, respectively. The average RSRs for models trained on recombining 2, 3, 4, and 5 proteins are 76.3%, 73.1%, 73.2%, and 78.9%, respectively. The second row shows the generated protein structures. We show the protein structures with the best and median scTM scores. The number under each protein structure is the scTM score.

| Length | 100 | 120 | 140 | 160 | 180 | 200 | 220 | 240 | 260 | AVG |
|---|---|---|---|---|---|---|---|---|---|---|
| FrameDiff | 36.00 | 48.00 | 56.00 | 59.00 | 50.00 | 47.00 | 58.59 | 64.00 | 65.30 | 53.77 |
| VFNDiff | 36.00 | 44.33 | 51.00 | 54.55 | 55.56 | 58.16 | 52.92 | 60.42 | 62.00 | 52.77 |
| FrameFlow | 41.41 | 52.04 | 59.00 | 62.24 | 58.59 | 62.63 | 57.73 | 63.27 | 65.80 | 58.08 |
| RFM(2) | 71.43 | 69.81 | 62.96 | 66.67 | 71.70 | 74.07 | 63.64 | 75.86 | 79.20 | **70.59** |
| RFM(3) | 88.89 | 74.19 | 75.61 | 75.61 | 75.41 | 75.00 | 72.00 | 80.36 | 81.80 | **77.65** |

Table 1: Novelty (%) comparison across different models. RFM(2) and RFM(3) indicate the generation is obtained through recombining 2 and 3 different proteins respectively. AVG indicates the average of novelty over all lengths.

respectively. The other hyperparameter setting of RFM follows (Yim et al., 2024). More details can be found in the Appendix.A.2.2

## 5.2 EXPERIMENTAL RESULTS

We first assess the recombination ability of RFM using the recombination success rate. Next, we compare RFM with other methods on the task of generating novel proteins and evaluate their performance using novelty metrics. Finally, we demonstrate the generalizability of RFM through experiments involving the recombination of different numbers of proteins.

### 5.2.1 RECOMBINATION ABILITY OF RFM

We demonstrate the recombination ability of RFM by training four models on the task of recombining two, three, four, and five proteins. The scTM scores of these models on the task of recombining two proteins is shown in the first row of Fig. 3. The generation with a scTM score over 0.5 is considered successful. The average RSRs for models trained on recombining two, three, four, and five proteins are 76.3%, 73.1%, 73.2%, and 78.9%, respectively. From these results, we have made the following observations: (1) Although the RSRs of the models trained on recombining three and four proteins are slightly lower than the model trained on recombining two proteins, the RSRs of all models are above 70%, indicating the effectiveness of RFM in recombining different numbers of proteins. (2) The distribution of scTM scores from the models trained on recombining different numbers of proteins are similar, which demonstrates the generalization and recombination ability of RFM. (3) The RSRs vary with the length of the protein, which is due to the recombination flexibility.

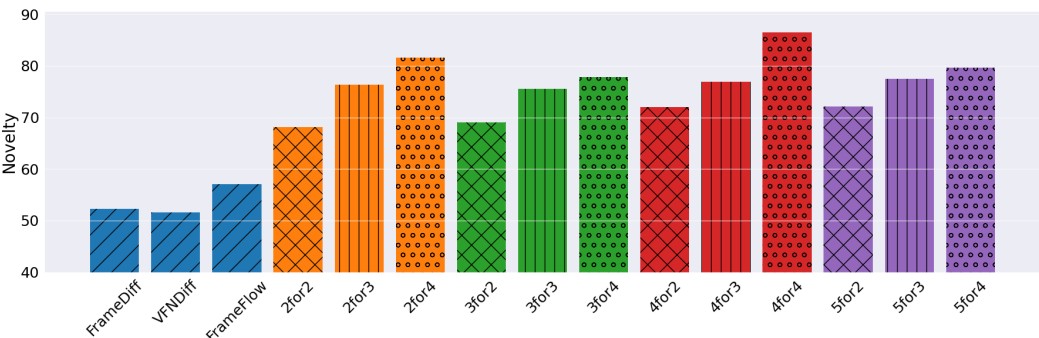

Figure 4: Novelty of generated proteins from different methods. The horizontal and vertical coordinates are the methods and novelty, respectively. The method with name $n$for$m$ indicates the model is trained on recombining $n$ proteins and tested on recombining $m$ proteins.

| # proteins | 2 | 3 | 4 | 5 |
|---|---|---|---|---|
| $[100, 260]$ | 73.16 | 43.00 | 16.00 | 7.50 |
| $(260, 400]$ | 67.73 | 51.70 | 37.92 | 27.50 |
| Average | 71.31 | 47.35 | 26.96 | 17.50 |
| Top-5 | 79.38 | 56.74 | 43.00 | 33.33 |

Table 2: RSR(%) of RFM on recombining various numbers of proteins with various length.

| # proteins | 2 | 3 | 4 | 5 |
|---|---|---|---|---|
| $[100, 260]$ | 72.04 | 76.90 | 86.50 | 73.89 |
| $(260, 400]$ | 65.49 | 80.84 | 73.01 | 75.92 |
| Average | 68.76 | 78.87 | 79.75 | 74.90 |
| Top-5 | 78.69 | 84.84 | 78.65 | 81.02 |

Table 3: Novelty(%) of RFM on recombining various numbers of proteins with various length.

For proteins with more residues, more free residues are available, which makes the recombination easier and more flexible.

To provide a visual representation, we showcase the protein structures with the best and median scTM scores in the second row of Fig. 3. The number under each protein structure is the scTM score. These results demonstrate the ability of RFM to effectively generate proteins, regardless of the number of proteins involved in the recombination task.

### 5.2.2 NOVEL PROTEIN GENERATION

We compare RFM with other unconditional protein generation methods on the task of generating novel proteins. The novelty of the generated proteins is evaluated using the pdbTM score. The results are shown in Table. 1. The average novelty of RFM in recombining two and three proteins is 68.11% and 77.19% respectively, which is 19.11% higher than the previous best method, FrameFlow. These results demonstrate the effectiveness of RFM in generating novel proteins. With more proteins involved in the recombination, RFM achieves a higher novelty since it is more likely to get segments that never exist in nature.

### 5.2.3 GENERALIZATION OF RFM

To demonstrate the generalization of RFM, we first evaluate its recombination ability to recombine different numbers of proteins and generate proteins with lengths out of the training scope. Then we measure the generation novelty of each method for recombining different numbers of proteins.

The RSRs of RFM on recombining various numbers of proteins with different lengths are shown in Table. 2. We have the following observations: (1) Although the RSRs of the RFM were higher at length [100,260] than at length (260,400], which is out of the training scope, they both achieved high RSRs of more than 65%. This demonstrates the generalization of RFM in recombining proteins at different lengths. (2) The RSR of recombining three, four, and five proteins are lower within the sequence length range of [100, 260]. The RSR decreases as the number of proteins involved in recombination increases. This trend occurs because fewer free residues remain available for recombining the selected segments. (3) The top-5 RSRs of RFM on recombining two, three, four,

| Recombination | Movable | 100 | 140 | 180 | 220 | 260 | Average |
|---|---|---|---|---|---|---|---|
| | | 41.41 | 59.00 | 58.59 | 57.73 | 65.80 | 58.08 |
| ✓ | | 68.42 | 67.86 | 68.00 | 67.86 | 76.79 | 68.27 |
| ✓ | ✓ | 71.43 | 62.96 | 71.70 | 63.64 | 79.20 | 70.59 |

Table 4: Ablation study of RFM on novelty (%) of generated proteins. Recombination and Movable indicate the recombination process and making the recombination segments movable respectively.

and five proteins are 79.38%, 56.74%, 43.00%, and 33.33% respectively. The results demonstrate the effectiveness of RFM in recombining different numbers of proteins.

The novelty of RFM on recombining various numbers of proteins are shown in Fig. 4. Even trained on recombining different numbers of proteins, the novelty on recombining various numbers of proteins remains high outperforming previous unconditional generation methods, which demonstrates the generalization of RFM on recombining various number of proteins. In addition, novelty increases as the number of proteins involved in recombination increases, due to the higher probability of combining protein fragments that have never existed together in nature. The novelty of generation proteins at various lengths are high as shown in Table. 3. Especially for proteins with a length of (260, 400], which is out of the training scope, the novelty of RFM is still high, which demonstrates the generalization of RFM in generating novel proteins at different lengths.

### 5.3 ABLATION STUDY

We evaluate the effectiveness of the recombination process and making the recombination segments movable. Without the recombination process, the model generates proteins by directly sampling from the prior distribution. Without both the Recombination and the Movable, the model performs much worse than the RFM. Although the model without the Movable is comparable to the RFM, the average RSR is much lower than the RFM (73.20% V.S. 46.60%). More results can be found in Appendix. A.4.2.

## 6 DISCUSSION

**How does RFM generate novel proteins?**  RFM generates novel proteins by recombining different proteins and making the recombination segments movable. The recombination process ensures the existence of selected recombination segments in the generation. The movable process allows the recombination segments to move freely in the generation. Attributed to the recombination process and the movable process, protein segments that have never existed together in nature can be combined to generate novel proteins.

**How does RFM generalize to recombining various proteins at various lengths?**  The generalization of RFM is attributed to the fact that all residues are treated equally in the flow matching. The only difference is that we average the rotation and translation of the selected recombination segments. This design allows RFM to generalize to recombining various proteins at various lengths. More details can be found in Appendix. A.1.4.

## 7 CONCLUSION

In this work, we propose a novel protein generation model, Recombination Flow Matching (RFM), which generates novel proteins by recombining different proteins. This concept is inspired by natural protein evolution, where recombination plays a crucial role in the emergence of novel proteins. To integrate recombination within the flow matching model, we employ rigid-body dynamics to preserve the structural integrity of the selected recombination segments while allowing for their mobility. RFM demonstrates high success rates and novelty in recombining varying numbers of proteins and producing proteins across different lengths, showcasing its effectiveness and generalizability in novel protein generation. In future work, we aim to enhance the model by automating and increasing the flexibility of recombination segment selection.

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

# A  APPENDIX

## A.1  METHOD DETAILS

### A.1.1  PROTEIN PARAMETERIZATION

**Gram-Schmidt process.**  The Gram-Schmidt process is an algorithm in linear algebra that takes a set of linearly independent vectors and transforms them into a set of orthogonal (mutually perpendicular) vectors while maintaining the same span (subspace). Given two vectors $v_C$ and $v_N$ as shown in Fig. 1(left), the goal is to convert them into an orthogonal vector $r$, such that the vectors $v_C$ and $v_N$ are orthogonal to the vector $r$. Steps of Gram-Schmidt:

1. Get the normalized $\boldsymbol{v}_C$ and $\boldsymbol{v}_N$: $\boldsymbol{u}_1 = \boldsymbol{v}_C/\|\boldsymbol{v}_C\|$, $\boldsymbol{u}_2 = \boldsymbol{v}_N/\|\boldsymbol{v}_N\|$.

2. Get the orthogonal vector of $\boldsymbol{v}_C$ and $\boldsymbol{v}_N$: $\boldsymbol{u}_3 = \boldsymbol{v}_C \times \boldsymbol{v}_N$.

3. Get the rotation matrix for the residue: $\boldsymbol{R} = concat(\boldsymbol{u}_1, \boldsymbol{u}_2, \boldsymbol{u}_3)$

### A.1.2 Flow Definition

Following Yim et al. (2024), the time derivative of the rotation and translation manifolds are defined as:

$$\dot{\boldsymbol{x}}_t^{(n)} = \frac{\boldsymbol{x}_1^{(n)} - \boldsymbol{x}_t^{(n)}}{1 - t}$$

$$\dot{\boldsymbol{R}}_t^{(n)} = \frac{\log_{\boldsymbol{R}_t^{(n)}}(\boldsymbol{R}_1^{(n)})}{1 - t}$$

$$v_{\boldsymbol{x}}^{(n)} := \frac{\hat{\boldsymbol{x}}_1^{(n)} - \boldsymbol{x}_t^{(n)}}{1 - t}$$

$$v_{\boldsymbol{R}}^{(n)} := \frac{\log_{\boldsymbol{R}_t^{(n)}}(\hat{\boldsymbol{R}}_1^{(n)})}{1 - t}$$

### A.1.3 Derivation of Rigid Dynamics

We derive the rigid dynamics involved in the RFM in this section.

Conditional Flow

**Rotation manifold.** Given the rotation at $t = t$, we have $R_t = \exp_{R_0}(t \log_{R_0} R_1)$. The rotation is simplified as follows:

$$R_t = R_1^t \tag{18}$$

We first transform the rotation into the virtual coordinate system as $R_t' = \boldsymbol{I}R_t$. Then we calculate the transformation between $R_t$ and $R_0$ in the virtual coordinate system as follows:

$$\Delta R = (\boldsymbol{I}R_t)(\boldsymbol{I}R_0)^{-1}$$
$$= R_t R_0^{-1}$$

Then we average the rotation of the selected recombination segments and apply it to each residue as follows:

$$R_t = \left[ \left( \overline{R_t R_0^{-1}} \right)_s, \left( R_t R_0^{-1} \right)_r \right] R_0$$
$$= \left[ \left( \overline{R_t R_0^{-1}} R_0 \right)_s, \left( R_t \right)_r \right]$$

Since the rotation of rigid is accompanied by translation, we calculate the translation of the residue accompanying its rotation as follows: First of all, we transform the translation of the residues into the virtual coordinate system as $\boldsymbol{X}' = \boldsymbol{X} - \boldsymbol{o}$. Then the transformation of translation in the virtual coordinate system is calculated as:

$$\boldsymbol{X}'' = \left[ \left( \overline{R_t R_0^{-1}} \right)_s, \left( R_t R_0^{-1} \right)_r \right] \boldsymbol{X}'$$
$$= \left[ \left( \overline{R_t R_0^{-1}} \right)_s, \left( R_t R_0^{-1} \right)_r \right] (\boldsymbol{X} - \boldsymbol{o})$$

Next, we transform the translation back to the original coordinate system and calculate the translation transformation. Since for the residues that are not in the selected recombination segments. $\boldsymbol{o} = \boldsymbol{x}$. We have:

$$\Delta \boldsymbol{X} = \left[ \left( \overline{R_t R_0^{-1}} \right)_s, \left( R_t R_0^{-1} \right)_r \right] (\boldsymbol{X} - \boldsymbol{o}) + \boldsymbol{o} - \boldsymbol{X}$$
$$= \left[ \left( \overline{R_t R_0^{-1}(\boldsymbol{X} - \boldsymbol{o})} \right)_s, \left( R_t R_0^{-1}(\boldsymbol{X} - \boldsymbol{o}) \right)_r \right] + \boldsymbol{o} - \boldsymbol{X}$$
$$= \left[ \left( \overline{R_t R_0^{-1}(\boldsymbol{X} - \boldsymbol{o})} + \boldsymbol{o} - \boldsymbol{X} \right)_s, 0_r \right]$$

**Translation manifold.** We average the translation of the selected recombination segments. To maintain the structural integrity of the segments being recombined, we add the translation accompanied by the rotation to the translation transformation as follows:

$$\begin{aligned}
\boldsymbol{X}_t &= t(\boldsymbol{X}_1 - \boldsymbol{X}_0) + \boldsymbol{X}_0 + \Delta\boldsymbol{X} \\
&= [t(\boldsymbol{X}_1 - \boldsymbol{X}_0), 0_r] + \boldsymbol{X}_0 + \Delta\boldsymbol{X} \\
&= \left[\overline{t(\boldsymbol{X}_{s1} - \boldsymbol{X}_{s0})} + \boldsymbol{X}_{s0}, \quad t(\boldsymbol{X}_{r1} - \boldsymbol{X}_{r0}) + \boldsymbol{X}_{r0}\right] + \Delta\boldsymbol{X}
\end{aligned}$$

Model Architecture

**Rotation manifold.** Given the update prediction for rotation $R^{(l-1)}$ as $\hat{R}_\delta$. We first apply the update prediction to the rotation manifold, then we average the rotation of the selected recombination segments at the virtual coordinate system and apply it to each residue as follows:

$$\begin{aligned}
R^{(l)} &= \hat{R}_\delta R^{(l-1)} \\
&= (I^{-1}(\hat{R}_\delta R^{(l-1)})(I^{-1}R^{(l-1)})^{-1})R^{(l-1)} \\
&= \hat{R}_\delta R^{(l-1)} \\
&= \left[\left(\overline{\hat{R}_\delta}\right)_s, \quad \left(\hat{R}_\delta\right)_r\right] R^{(l-1)}
\end{aligned}$$

The accompanied translation is calculated as follows: First of all, we transform the translation of the residues into the virtual coordinate system as $\boldsymbol{X}^{(l-1)'} = \boldsymbol{X}^{(l-1)} - \boldsymbol{o}$. Then, The transformation of translation in the virtual coordinate system is calculated as:

$$\begin{aligned}
\boldsymbol{X}^{(l-1)''} &= \left[\left(\overline{\hat{R}_\delta}\right)_s, \quad \left(\hat{R}_\delta\right)_r\right] \boldsymbol{X}^{(l-1)'} \\
&= \left[\left(\overline{\hat{R}_\delta}\right)_s, \quad \left(\hat{R}_\delta\right)_r\right] (\boldsymbol{X}^{(l-1)} - \boldsymbol{o})
\end{aligned}$$

Next, we transform the translation back to the original coordinate system and calculate the translation transformation. Since for the residues that are not in the selected recombination segments. $\boldsymbol{o} = \boldsymbol{x}^{(l-1)}$. We have:

$$\begin{aligned}
\Delta\boldsymbol{X}^{(l-1)} &= \left[\left(\overline{\hat{R}_\delta}\right)_s, \quad \left(\hat{R}_\delta\right)_r\right] (\boldsymbol{X}^{(l-1)} - \boldsymbol{o}) + \boldsymbol{o} - \boldsymbol{X}^{(l-1)} \\
&= \left[\left(\overline{\hat{R}_\delta(\boldsymbol{X}^{(l-1)} - \boldsymbol{o})}\right)_s, \quad \left(\hat{R}_\delta(\boldsymbol{X}^{(l-1)} - \boldsymbol{o})\right)_r\right] + \boldsymbol{o} - \boldsymbol{X}^{(l-1)} \\
&= \left[\left(\overline{\hat{R}_\delta(\boldsymbol{X}^{(l-1)} - \boldsymbol{o})} + \boldsymbol{o} - \boldsymbol{X}^{(l-1)}\right)_s, \quad 0_r\right]
\end{aligned}$$

**Translation manifold.** We average the translation of the selected recombination segments. To maintain the structural integrity of the segments being recombined, we add the translation accompanied by the rotation to the translation transformation as follows:

$$\begin{aligned}
\boldsymbol{X}_t &= (R^{(l-1)}\hat{\boldsymbol{X}}_\delta) + \boldsymbol{X}^{(l-1)} + \Delta\boldsymbol{X}^{(l-1)} \\
&= \left[\left(\overline{R^{(l-1)}\hat{\boldsymbol{X}}_\delta}\right)_s, R^{(l-1)}\hat{\boldsymbol{X}}_\delta\right] + \boldsymbol{X}^{(l-1)} + \left[\left(\overline{\hat{R}_\delta(\boldsymbol{X}^{(l-1)} - \boldsymbol{o})} + \boldsymbol{o} - \boldsymbol{X}^{(l-1)}\right)_s, \quad 0_r\right] \\
&= \left[\left(\overline{\hat{R}_\delta(\boldsymbol{X}^{(l-1)} - \boldsymbol{o})} + \boldsymbol{o} - \boldsymbol{X}^{(l-1)} + \overline{R^{(l-1)}\hat{\boldsymbol{X}}_\delta}\right)_s, \left(R^{(l-1)}\hat{\boldsymbol{X}}_\delta\right)_r\right] + \boldsymbol{X}^{(l-1)}
\end{aligned}$$

Inference

**Rotation manifold.** Given the predicted rotation $\hat{R}$ and previous rotation $R^{(t-1)}$, we can calculate transformation between $\hat{R}$ and $R^{(t-1)}$ in the virtual coordinate system as follows:

$$\Delta R = \hat{R}(R^{(t-1)})^{-1}\Delta t/(1 - t)$$

Then we average the rotation transformation of the selected recombination segments and apply it to each residue as follows:

$$R^{(t)} = \hat{R}(R^{(t-1)})^{-1}\Delta t/(1-t)R^{(t-1)}$$

$$= \left[(\Delta R)_s, \quad \left(\hat{R}(R^{(t-1)})^{-1}\Delta t/(1-t)\right)_r\right]R^{(t-1)}$$

The accompanied translation is calculated the same as the conditional flow as follows:

$$\boldsymbol{X}^{(t-1)''} = \left[(\Delta R)_s, \quad \left(\hat{R}(R^{(t-1)})^{-1}\Delta t/(1-t)\right)_r\right](\boldsymbol{X}^{(t-1)} - \boldsymbol{o})$$

$$\Delta \boldsymbol{X}^{(t-1)} = \left[(\Delta R)_s, \quad \left(\hat{R}(R^{(t-1)})^{-1}\Delta t/(1-t)\right)_r\right](\boldsymbol{X}^{(t-1)} - \boldsymbol{o}) + \boldsymbol{o} - \boldsymbol{X}^{(t-1)}$$

$$= \left[\left(\Delta R(\boldsymbol{X}^{(t-1)} - \boldsymbol{o}) + \boldsymbol{o} - \boldsymbol{X}^{(t-1)}\right)_s, 0_r\right] + \boldsymbol{X}^{(t-1)}$$

**Translation manifold.** We average the translation of the selected recombination segments. To maintain the structural integrity of the segments being recombined, we add the translation accompanied by the rotation to the translation transformation as follows:

$$\boldsymbol{X}^{(t)} = \frac{(\hat{\boldsymbol{X}} - \boldsymbol{X}^{(t-1)})\Delta t}{1-t} + \boldsymbol{X}^{(t-1)} + \Delta \boldsymbol{X}^{(t-1)}$$

$$= \left[\left(\frac{(\hat{\boldsymbol{X}} - \boldsymbol{X}^{(t-1)})\Delta t}{1-t}\right)_s, \left(\frac{(\hat{\boldsymbol{X}} - \boldsymbol{X}^{(t-1)})\Delta t}{1-t}\right)_r\right] + \boldsymbol{X}^{(t-1)} + \Delta \boldsymbol{X}^{(t-1)}$$

$$= \left[\left(\Delta R(\boldsymbol{X}^{(t-1)} - \boldsymbol{o}) + \boldsymbol{o} - \boldsymbol{X}^{(t-1)} + \overline{\frac{(\hat{\boldsymbol{X}} - \boldsymbol{X}^{(t-1)})\Delta t}{1-t}}\right)_s, \left(\frac{(\hat{\boldsymbol{X}} - \boldsymbol{X}^{(t-1)})\Delta t}{1-t}\right)_r\right]$$

$$+ \boldsymbol{X}^{(t-1)}$$

### A.1.4 RELATIONSHIP BETWEEN RFM AND FM

We analyze the relationship between recombination flowing matching(RFM) and flow matching(FM). RFM and FM are consistent since we only average the rotation and translation of the selected recombination segments. By removing the average operation, we find the operations on all the residues are the same.

**Conditional flow.** For Eq. 9, we have

$$R_t = \left[\overline{R_{s1}^t R_{s0}^{-1}}R_{s0}, \quad R_{r1}^t\right]$$

$$= \left[R_{s1}^t R_{s0}^{-1} R_{s0}, \quad R_{r1}^t\right]$$

$$= \left[R_{s1}^t, \quad R_{r1}^t\right] = R_1^t.$$

For Eq. 12, if we remove the average operation, since $\boldsymbol{o} = \overline{\boldsymbol{X}}$,, i.e., $\boldsymbol{o} - \boldsymbol{X} = 0$, we have:

$$\Delta \boldsymbol{X} = [\Delta R_s(\boldsymbol{X}_s - \boldsymbol{o}) + \boldsymbol{o} - \boldsymbol{X}_s, \quad 0_r]$$

$$= 0$$

For Eq. 13, we further remove the average operation on the translation prediction, then we have:

$$\boldsymbol{X}_t = \left[\overline{t(\boldsymbol{X}_{s1} - \boldsymbol{X}_{s0})} + \boldsymbol{X}_{s0}, \quad t(\boldsymbol{X}_{r1} - \boldsymbol{X}_{r0}) + \boldsymbol{X}_{r0}\right] + \Delta \boldsymbol{X}$$

$$= [t(\boldsymbol{X}_{s1} - \boldsymbol{X}_{s0}) + \boldsymbol{X}_{s0}, \quad t(\boldsymbol{X}_{r1} - \boldsymbol{X}_{r0}) + \boldsymbol{X}_{r0}] + 0$$

$$= t(\boldsymbol{X}_1 - \boldsymbol{X}_0) + \boldsymbol{X}_0$$

**Model architecture.** For Eq. 14

$$R^{(l)} = \left[\left(\overline{\hat{R}_\delta}\right)_s, \quad \left(\hat{R}_\delta\right)_r\right]R^{(l-1)}$$

$$= \left[\left(\hat{R}_\delta\right)_s, \quad \left(\hat{R}_\delta\right)_r\right]R^{(l-1)}$$

$$= \hat{R}_\delta R^{(l-1)}$$

For Eq. 15, since $\boldsymbol{o} = \overline{\boldsymbol{X}^{(l-1)}}$, $i.e.$, $\boldsymbol{o} - \boldsymbol{X}^{(l-1)} = 0$ we have:

$$
\begin{aligned}
\boldsymbol{X}^{(l)} &= \left[ \left( \hat{R}_\delta (\boldsymbol{X}^{(l-1)} - \boldsymbol{o}) + \boldsymbol{o} - \boldsymbol{X}^{(l-1)} + R^{(l-1)} \hat{\boldsymbol{X}}_\delta \right)_s , \left( R^{(l-1)} \hat{\boldsymbol{X}}_\delta \right)_r \right] + \boldsymbol{X}^{(l-1)} \\
&= \left[ \left( 0 + R^{(l-1)} \hat{\boldsymbol{X}}_\delta \right)_s , \left( R^{(l-1)} \hat{\boldsymbol{X}}_\delta \right)_r \right] + \boldsymbol{X}^{(l-1)} \\
&= R^{(l-1)} \hat{\boldsymbol{X}}_\delta + \boldsymbol{X}^{(l-1)}
\end{aligned}
$$

**Inference.** For the rotation in inference, after removing the average operation, we have:

$$
\begin{aligned}
R^{(t)} &= \left[ \left( \hat{R}(R^{(t-1)})^{-1} \Delta t / (1-t) \right)_s , \quad \left( \hat{R}(R^{(t-1)})^{-1} \Delta t / (1-t) \right)_r \right] R^{(t-1)} \\
&= \left[ \left( \hat{R}(R^{(t-1)})^{-1} \Delta t / (1-t) \right)_s , \quad \left( \hat{R}(R^{(t-1)})^{-1} \Delta t / (1-t) \right)_r \right] R^{(t-1)} \\
&= \hat{R}(R^{(t-1)})^{-1} \Delta t / (1-t) R^{(t-1)}
\end{aligned}
$$

For the translation in inference, we remove the average operation and we have $\boldsymbol{o} = \overline{\boldsymbol{X}^{(t-1)}}$, ie, $\boldsymbol{o} - \boldsymbol{X}^{(t-1)} = 0$:

$$
\begin{aligned}
\boldsymbol{X}^{(t)} &= \left[ \left( \Delta R(\boldsymbol{X}^{(t-1)} - \boldsymbol{o}) + \boldsymbol{o} - \boldsymbol{X}^{(t-1)} + \frac{\overline{(\hat{\boldsymbol{X}} - \boldsymbol{X}^{(t-1)}) \Delta t}}{1-t} \right)_s , \left( \frac{(\hat{\boldsymbol{X}} - \boldsymbol{X}^{(t-1)}) \Delta t}{1-t} \right)_r \right] + \boldsymbol{X}^{(t-1)} \\
&= \left[ \left( \Delta R(\boldsymbol{X}^{(t-1)} - \boldsymbol{o}) + \boldsymbol{o} - \boldsymbol{X}^{(t-1)} + \frac{(\hat{\boldsymbol{X}} - \boldsymbol{X}^{(t-1)}) \Delta t}{1-t} \right)_s , \left( \frac{(\hat{\boldsymbol{X}} - \boldsymbol{X}^{(t-1)}) \Delta t}{1-t} \right)_r \right] + \boldsymbol{X}^{(t-1)} \\
&= \left[ \left( 0 + \frac{(\hat{\boldsymbol{X}} - \boldsymbol{X}^{(t-1)}) \Delta t}{1-t} \right)_s , \left( \frac{(\hat{\boldsymbol{X}} - \boldsymbol{X}^{(t-1)}) \Delta t}{1-t} \right)_r \right] + \boldsymbol{X}^{(t-1)} \\
&= \frac{(\hat{\boldsymbol{X}} - \boldsymbol{X}^{(t-1)}) \Delta t}{1-t} + \boldsymbol{X}^{(t-1)}
\end{aligned}
$$

## A.2 TRAINING DETAILS

### A.2.1 AUXILIARY LOSS.

The auxiliary structure losses are shown in Eq. 19 and Eq. 20:

$$
\mathcal{L}_{atom} = \frac{1}{4} \sum_{i=1}^{n} \sum_{\boldsymbol{x} \in \Omega} \| \boldsymbol{x}_i - \hat{\boldsymbol{x}}_i \|^2 \tag{19}
$$

$$
\mathcal{L}_{pair} = \frac{1}{Z} \sum_{i,j=1}^{n} \sum_{a,b \in \Omega} \mathbb{1}\{d_{ab}^{ij} < 0.6\} \| d_{ab}^{ij} - \hat{d}_{ab}^{ij} \|^2, \tag{20}
$$

$$
Z = \left( \sum_{i,j=1}^{n} \sum_{a,b \in \Omega} \mathbb{1}\{d_{ab}^{ij} < 0.6\} \right) - n,
$$

where $\Omega$, $d_{ab}^{ij}$ and $\hat{d}_{ab}^{ij}$ are backbone atoms $\{\mathrm{C}, \mathrm{C}_\alpha, \mathrm{O}, \mathrm{N}\}$, the ground truth and predicted distance between atom respectively.

### A.2.2 SETUP

**Scheduler.** We use the scheduler of $\boldsymbol{k}(t) = 1 - t$ for training and $\boldsymbol{k}(t) = e^{-10t}$ for inference following (Yim et al., 2024).

**Hyperparameters.** We train the RFM with an optimizer of AdamW. The learning rate is 0.0001. All other settings of hyperparameter follow Yim et al. (2024).

**Hardware.** We train RFM for 2000 steps in around 14 hours. All our experiments are conducted on a computing cluster with 8 GPUs of NVIDIA GeForce RTX 4090 24GB and CPUs of AMD EPYC 7763 64-Core of 3.52GHz. All the inferences are conducted on a single GPU of NVIDIA GeForce RTX 4090 24GB.

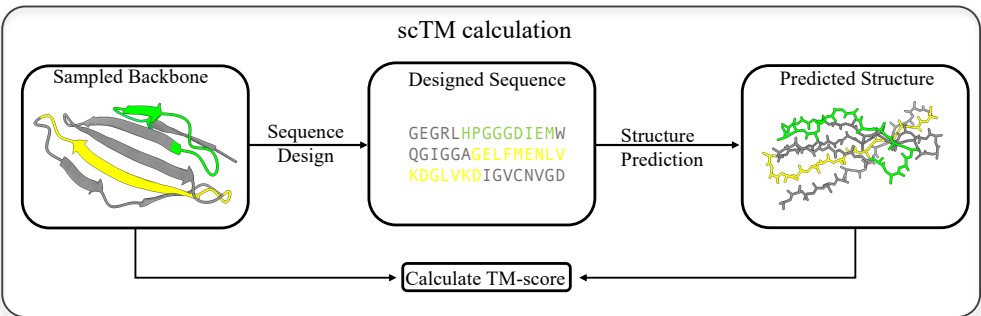

Figure 5: scTM calculation pipeline. The model for sequence design is proteinMPNN and the model for structure prediction is ESMFold. We use ProteinMPNN to generate the sequence from the provided protein structure. Next, the designed sequence is processed through ESMFold to reconstruct the full protein structure. Finally, we evaluate the similarity between the predicted and original backbone structures by calculating the TM-score.

### A.3 EXPERIMENT DETAILS

#### A.3.1 EVALUATION METRICS.

We mainly evaluate the models with the metrics of recombination success rate (RSR) and Novelty.

**RSR** is the percentage of designable proteins in the generation for our model. One protein is designable if scTM $> 0.5$. As shown in Fig. 5, the scTM score for a generated protein structure is computed in three steps following (Yim et al., 2023; Liu et al., 2024). First, we use ProteinMPNN (Dauparas et al., 2022) to design an amino acid sequence. Next, this sequence is processed by ESMFold (Lin et al., 2023) to predict the corresponding protein structure. Finally, the TM-score is calculated by comparing the predicted structure from ESMFold with the structure generated by our model. For each generated protein, the designed sequence from ProteinMPNN is input into ESMFold eight times to predict multiple folded structures. If any of the eight predicted structures has a TM-score greater than 0.5 when compared to the generated structure, the generation is deemed successful.

**Novelty.** One generation is novel if the TM-score with all the proteins in the PDB dataset (PDB-TM) is below 0.7 following previous works (Yim et al., 2023; Mao et al., 2024; Yim et al., 2024). The novelty is the percentage of novel proteins in the generated designable proteins. We generate 100 samples for each length and report their novelty.

**Diversity.** We also use the MaxCluster (Herbert & Sternberg, 2008) to calculate the diversity of generation. Diversity is the ratio of unique clusters in the number of generated samples where the clusters are produced by MaxCluster following previous works. The diversity of generation for RFM is always 1.

### A.4 ADDITIONAL EXPERIMENTAL RESULTS

#### A.4.1 RECOMBINATION SUCCESS RATE

The scTM scores of models trained on recombining different numbers of proteins and tested on recombining different numbers of proteins are shown in Fig. 6. We have the following observations: (1) The RSR distribution of models trained and tested on recombining different numbers of proteins are similar, demonstrating the generalization of RFM. (2) With the length of the generated protein increasing, the RSR increases. While the RSR decreases, the number of proteins to be recombined increases. The reason for this observation is that with more free residues in the generation, the recombination is much easier.

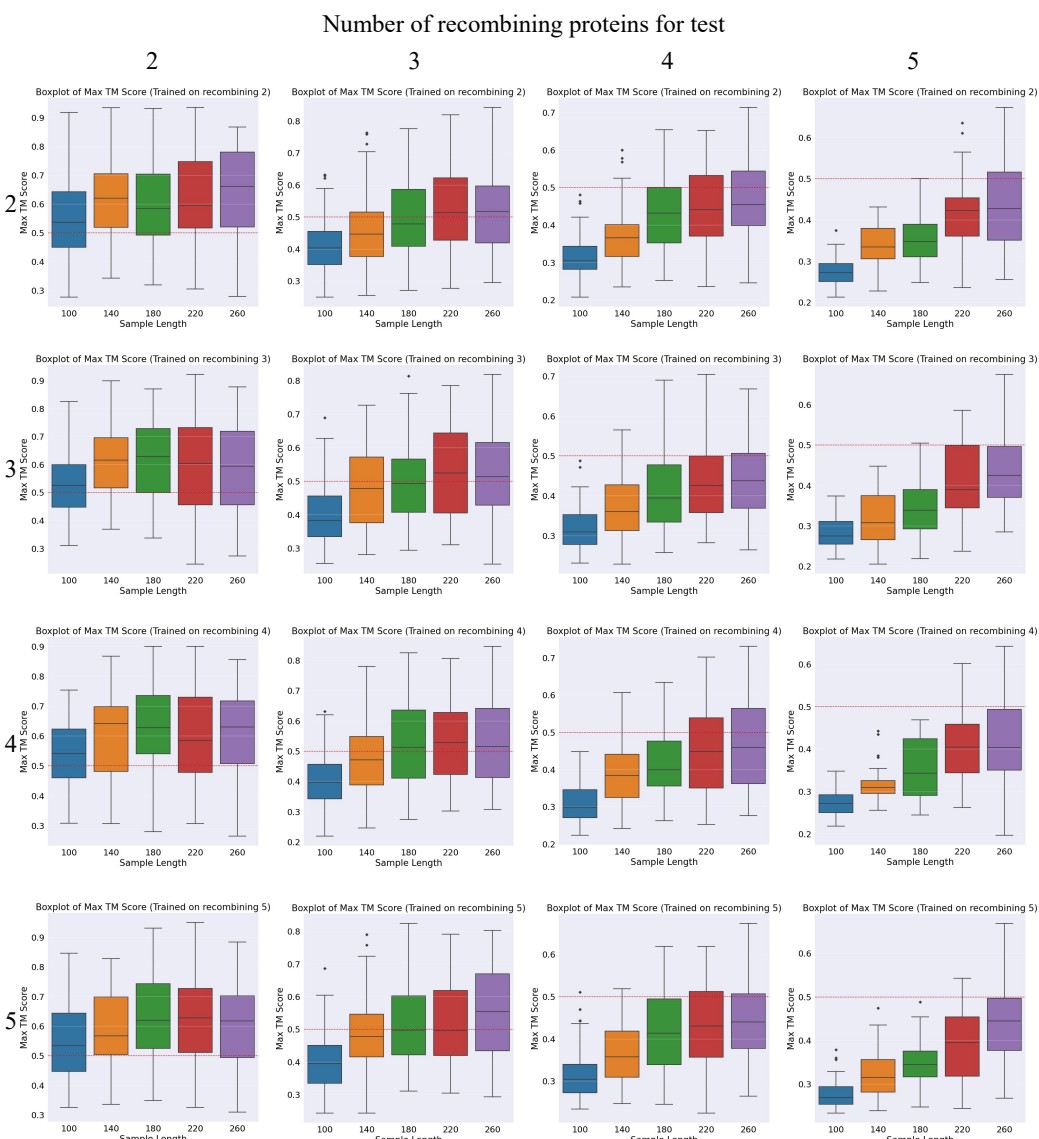

Figure 6: The recombination results of training and testing RFM on recombining different numbers of proteins. The horizontal and vertical coordinates in each subfigure are the length of the protein and the scTM score, respectively.

### A.4.2 ABLATION STUDY

The model without making the selected recombination segments movable is trained like inpainting. We randomly select 2 segments of lengths between 2 and 40 residues from one protein. Then the model is trained to outpaint the other residues. We show the RSR of the two models in Table. 5.

| MV | 100 | 120 | 140 | 160 | 180 | 200 | 220 | 240 | 260 | Average |
|----|-----|-----|-----|-----|-----|-----|-----|-----|-----|---------|
|    | 49.23 | 80.0 | 80.0 | 72.31 | 76.92 | 78.46 | 63.08 | 80.0 | 78.46 | 73.16 |
| ✓  | 23.00 | 31.00 | 41.00 | 41.00 | 53.00 | 54.00 | 55.00 | 56.00 | 65.38 | 46.60 |

Table 5: Ablation study of RFM on RSR (%) of generated proteins. MV indicates making the recombination segments movable.

### A.4.3 NOVELTY

To assess the impact of fine-tuning on the flow matching model, we also use the fine-tuned RFM to generate proteins unconditionally. The novelty of generation is shown in Table. 6. Even for unconditional generation, the novelty of generation increases.

| Length | 100 | 120 | 140 | 160 | 180 | 200 | 220 | 240 | 260 | AVG |
|--------|-----|-----|-----|-----|-----|-----|-----|-----|-----|-----|
| FrameDiff | 36.00 | 48.00 | 56.00 | 59.00 | 50.00 | 47.00 | 58.59 | 64.00 | 65.30 | 53.77 |
| VFNDiff | 36.00 | 44.33 | 51.00 | 54.55 | 55.56 | 58.16 | 52.92 | 60.42 | 62.00 | 52.77 |
| FrameFlow | 41.41 | 52.04 | 59.00 | 62.24 | 58.59 | 62.63 | 57.73 | 63.27 | 65.80 | 58.08 |
| RFM(2) | 71.43 | 69.81 | 62.96 | 66.67 | 71.70 | 74.07 | 63.64 | 75.86 | 79.20 | **70.59** |
| RFM(3) | 88.89 | 74.19 | 75.61 | 75.61 | 75.41 | 75.00 | 72.00 | 80.36 | 81.80 | **77.65** |
| uncondition | 55.56 | 64.29 | 59.26 | 62.07 | 68.97 | 64.00 | 62.07 | 81.48 | 80.40 | 66.45 |

Table 6: Novelty (%) comparison across different models.

### A.4.4 VISUAL RESUTLS

For each group of generation samples, we show the generated protein structures with the top-8 scTM score of each length as follows. We identify each selected recombination segment by the PDB identifier and the start and end sequence position on the protein.

**Recombine two proteins**   The generation results of recombining two proteins are shown below in Table. 7:

**Recombine three proteins**   The generation results of recombining three proteins are shown in Table. 8 below:

**Recombine four proteins**   The generation results of recombining four proteins are shown in Table. 9 below:

**Recombine five proteins**   The generation results of recombining five proteins are shown in Table. 10 below:

| | | |
|---|---|---|
| Structure | | |
| Protein | 6CTD_A_11_24&4RVB_A_292_301 | 6CTD_A_11_24&4RVB_A_292_301 |
| Length | 200 | 240 |
| scTM | 0.91 | 0.88 |
| Structure | | |
| Protein | 5Z2V_A_57_78&6CTD_A_11_24 | 6CTD_A_11_24&4RVB_A_292_301 |
| Length | 260 | 280 |
| scTM | 0.90 | 0.87 |
| Structure | | |
| Protein | 6CTD_A_11_24&
4RVB_A_292_301 | 3N0P_A_174_191&
4RVB_A_292_301 |
| Length | 300 | 320 |
| scTM | 0.87 | 0.87 |
| Structure | | |
| Protein | 5Z2V_A_57_78&1QIN_A_117_133 | 1QIN_A_117_133&1DQ1_A_85_94 |
| Length | 360 | 400 |
| scTM | 0.88 | 0.87 |

Table 7: Generation results for recombining two proteins.

| | | |
|---|---|---|
| Structure |  |  |
| Protein | 5Z2V_A_57_78&4RVB_A_292_301 &1DQ1_A_85_94 | 5Z2V_A_57_78&4RVB_A_292_301 &1DQ1_A_85_94 |
| Length | 240 | 260 |
| scTM | 0.82 | 0.82 |
| Structure |  |  |
| Protein | 5Z2V_A_57_78&4RVB_A_292_301 &1DQ1_A_85_94 | 5Z2V_A_57_78&4RVB_A_292_301 &1DQ1_A_85_94 |
| Length | 260 | 280 |
| scTM | 0.83 | 0.81 |
| Structure |  |  |
| Protein | 5Z2V_A_57_78&4RVB_A_292_301 &1DQ1_A_85_94 | 5Z2V_A_57_78&3N0P_A_174_191 &1DQ1_A_85_94 |
| Length | 320 | 360 |
| scTM | 0.82 | 0.80 |
| Structure |  |  |
| Protein | 3N0P_A_174_191&1QIN_A_117_133 &1DQ1_A_85_94 | 5Z2V_A_57_78&3N0P_A_174_191 &4RVB_A_292_301 |
| Length | 380 | 400 |
| scTM | 0.86 | 0.81 |

Table 8: Generation results for recombining three proteins.

| Structure |  |  |
|---|---|---|
| Protein | 5Z2V_A_57_78&6CTD_A_11_24 &4RVB_A_292_301&1DQ1_A_85_94 | 5Z2V_A_57_78&6CTD_A_11_24& 4RVB_A_292_301&1DQ1__A_85_94 |
| Length | 200 | 240 |
| scTM | 0.81 | 0.73 |
| Structure |  |  |
| Protein | 5Z2V_A_57_78&6CTD_A_11_24& 4RVB_A_292_301&1DQ1_A_85_94 | 5Z2V_A_57_78&3N0P_A_174_191 &4RVB_A_292_301&1DQ1_A_85_94 |
| Length | 260 | 300 |
| scTM | 0.79 | 0.79 |
| Structure |  |  |
| Protein | 5Z2V_A_57_78&1QIN_A_117_133 &6CTD_A_11_24&1DQ1_A_85_94 | 5Z2V_A_57_78&6CTD_A_11_24& 4RVB_A_292_301&1DQ1_A_85_94 |
| Length | 320 | 360 |
| scTM | 0.74 | 0.79 |
| Structure |  |  |
| Protein | 5Z2V_A_57_78&6CTD_A_11_24& 4RVB_A_292_301&1DQ1_A_85_94 | 5Z2V_A_57_78&1QIN_A_117_133 &6CTD_A_11_24&4RVB_A_292_301 |
| Length | 380 | 400 |
| scTM | 0.79 | 0.74 |

Table 9: Generation results for recombining four proteins.

| | | |
|---|---|---|
| Structure |  |  |
| Protein | 5Z2V_A_57_78&1QIN_A_117_133 &6CTD_A_11_24&4RVB_A_292_301 &1DQ1_A_85_94 | 5Z2V_A_57_78&3N0P_A_174_191& 1QIN_A_117_133&4RVB_A_292_301 &1DQ1_A_85_94 |
| Length | 240 | 260 |
| scTM | 0.61 | 0.67 |
| Structure |  |  |
| Protein | 3N0P_A_174_191&1QIN_A_117_133 &6CTD_A_11_24&4RVB_A_292_301 &1DQ1_A_85_94 | 5Z2V_A_57_78&1QIN_A_117_133& 6CTD_A_11_24&4RVB_A_292_301 &1DQ1_A_85_94 |
| Length | 300 | 320 |
| scTM | 0.63 | 0.75 |
| Structure |  |  |
| Protein | 3N0P_A_174_191&1QIN_A_117_133 &6CTD_A_11_24&4RVB_A_292_301 &1DQ1_A_85_94 | 5Z2V_A_57_78&3N0P_A_174_191 &6CTD_A_11_24&4RVB_A_292_301 &1DQ1_A_85_94 |
| Length | 340 | 360 |
| scTM | 0.72 | 0.72 |
| Structure |  |  |
| Protein | 5Z2V_A_57_78&1QIN_A_117_133 &6CTD_A_11_24&4RVB_A_292_301 &1DQ1_A_85_94 | 5Z2V_A_57_78&3N0P_A_174_191 &6CTD_A_11_24&4RVB_A_292_301 &1DQ1_A_85_94 |
| Length | 380 | 400 |
| scTM | 0.61 | 0.69 |

Table 10: Generation results for recombining five proteins.

