# OpenReview forum: "Recombination Flow Matching Model for Protein Evolution"
_ICLR.cc/2025/Conference — Submitted to ICLR 2025_

### Official Review · Reviewer_bYJt · 2024-10-21

**Soundness:** 3
**Presentation:** 1
**Contribution:** 2
**Rating:** 3
**Confidence:** 3

**Summary:**

The contribution is preserving the integrity of segments by using the same transformation across the selected segments (rigid dynamics). The experiment results show higher novelty score.

**Strengths:**

The experiments show 19.11% higher  in novelty score  than previous best method.

**Weaknesses:**

1. There are many imprecise definations of matrix dimensions, superscipt and subscript. Please see detailed list below:
   1. Line 132, the dimension of $T_i$ is $4\times4$, it cannot be decomposed to a $3\times3$ and a $3\times1$ in line 133.
   2. Eq.(1), $\mathsf{T}_i$ is  $4\times4$, it cannot be multiplied by a $3\times3$.
   3. Line 168, the subscript of $\mathsf{T}_t$ is defined in line 182 between (0,1),  contradicting to the subscript defined in line 151 which is the i-th residue.
   4. Eq.(8), the superscript of $R^{(t)}$ is time,  contradicting to the superscript defined Eq.(7) which is the n-th residue.
   5. Eq.(8), $t-1<0$ according to the interval of t defined in line 182 ($0<t<1$), which is meaningless for corresponding $R^{t<0}$.
   6. Line 237, the subscript of $\mathsf{T}_{sk}$ is defined as k-th segment, contradicting to the subscript defined in line 151 which is the i-th residue.
   7. Line 238, on the right side of the equation defining  $\mathsf{T}_{sk}$, it is independant of k, which should be dependant instead.
   8. Eq.(9), the time t appears both subscipt and superscript.
   9. Eq.(9), the superscript s1 represents the selected segment at time $t==1$, contradicting to the defination in line 237 which represents the 1st selected segment.
   10. Eq.(14), the superscript (l-1) represents layer, contradicting to the defination in line 263 which represents time (t-1).

2. The theoritical contribution (by using the same transformation across the selected segments) is not much. The authors may propose other techniques at the same paper. For example, similar to the two strategies in reference "Improved motif-scaffolding with SE(3) flow matching", 1) take the selected segments as input, 2) does not require training but bias the model's generative trajectory to preserve the selected segments.

3. There is no comparsion with other papers on designablility (or RSR). Please comprare with FrameFlow, FrameDiff and VFNDiff.

**Questions:**

1. For the metric of diversity, why can it achieve 1?



**Things to improve the paper that did not impact the score:**
1. Fig.1, $V_N, V_C, \psi$ are not defined when they occur at the first time, although some are defined in Appendix where they occur at the second time.
2. Line 317, precious works --> previous work.
3. Line 365, SRS is not defined.
4. Line 717 and 720, the two lines of derivations are the same.

---

### Official Review · Reviewer_aGU3 · 2024-10-30

**Soundness:** 1
**Presentation:** 2
**Contribution:** 2
**Rating:** 3
**Confidence:** 4

**Summary:**

Here, the authors introduce a new method for protein design that they claim is based upon biological principles of recombination, Recombination Flow Matching. Instead of designing proteins on a residue-level like recent protein generative model proposals, the method takes in larger fragments of protein backbones and transforms them with rotations and translations while keeping the structure rigid to design novel protein structures.

**Strengths:**

The authors introduce a clever way to train models that reconstruct plausible protein backbones from multiple fragments: during training, they propose breaking a single protein up into multiple fragments and training the model to reconstruct its original PDB structure, which effectively serves as the ground truth. The samples show that this training convincingly generalizes to inference, where they now use fragments from different proteins.

**Weaknesses:**

Major comments:
1) One of the central claims of the paper is that despite its importance in evolution, recombination is unexplored for protein design methods, and that their method, RFM, is novel as the first model to leverage the critical biological phenomena of recombination into protein design. However, the very premise of this claim falls short to me: while it is true that recombination is critical for evolution on a genome-level, there is much less evidence to suggest that it is one of the major drivers of diversification at the protein level. For example, in Kummerfield and Teichmann 2009, a global analysis of 192 genomes indicates that the linear order of domains in proteins is largely preserved across evolution, and re-arrangements are rare. Across species, the substitution rate greatly outstrips the recombination rate for proteins (see Arenas 2021). The authors provide one citation to back up their claim that recombination is critical to protein evolution (Netzer & Hartl 1997) - but this citation does not even establish this, but instead proposes sequential domain folding as a means to preserve the viability of protein products if recombination does result in novel combinations of domains, without establishing that this is indeed a widespread phenomenon. Considering how much the authors' claim of designing a novel method relies upon their claim of identifying a critical mechanism of evolution that has been so-far underlooked in protein design, the authors should actually establish that recombination at the protein level is indeed a significant driver of protein evolution.

2) Related to the last point, this manuscript conflates the term "recombination" as used by protein engineers, which generally refers to using fragments from existing proteins to design new ones, versus "recombination" as used by evolutionary biologists, which refers to a molecular mechanism. The problem is that the authors rely upon a claim that they're exploiting the latter definition to claim they're doing something novel, whereas they're largely achieving the former definition, which has numerous established papers. Despite claiming to "leverage recombination to generate novel protein structures", the method cannot be said to be even simulating recombination from a biological perspective. For example, there are no species constraints on pools of proteins fragments can be sampled from, no genomic constraints on chromosomal position or sequence constraints on homology, no probabilistic components making double recombination events swapping out just a few hundred KB much rarer compared to swapping out all of a protein sequence past a certain point. Instead, their method more aligns with the former definition used by protein engineers. From this perspective, the authors are missing an entire canon of related works by only focusing on recent generative models - recombining existing protein fragments was a common way to generate functional diversity prior to deep-learning generative models. Bedbrook et al. 2017/2019 design channelrhodopsins assisted by machine learning models this way. Romero et al 2012 purpose active learning algorithms to select sequence fragments from existing libraries to design enzymes. Voigt et al 2002 describes computational algorithms for this purpose. The claim that RFM is the first model that leverages "recombination", at least from a protein engineering perspective, is not true - this is a well-established strategy dating back decades.

3) Especially because this method bears similarity to existing protein engineering methods, the evaluation metrics the authors are using are insufficient. First, the authors use a pdbTM score to claim that the model is generating novel structures. However, it's important for them to keep in mind the point of a novelty evaluation, which is to support the claim that the structures generated by a generative method reflect novel functions beyond proteins present in PDB. Recombination-based design methods generally have the limitation that because they're using templates from existing proteins, it is unlikely that substantially different functions will emerge. Hence, this metric is easily gamed - if I took two protein sequences, chopped each in half, and glued them together, the pdbTM will technically be dissimilar from either source protein, but it's unlikely I'm covering new ground in function space, because I'm just recycling domains from these proteins. Secondly, the scTM score is biased by the fact that the authors are using relatively large chunks of backbones from existing PDB structures. ProteinMPNN itself is trained on PDB, so this raises the question of if scTM is actually a good proxy for designability, or if the model has simply overfit to structure-sequence pairs already seen during training. To generalize these comments - the authors should consider how to adapt metrics recognizing the limitations of their specific method, instead of simply recycling routine protein generation metrics, and I would heavily encourage them to look at the prior literature of recombination-based methods in protein design for this purpose.

Minor comments:
1) Abstract - "current methodologies often face significant limitations in generating both novel protein structures" -> both novel and what? This sentence doesn't parse.

**Questions:**

See weaknesses

---

### Official Review · Reviewer_eGSF · 2024-10-31

**Soundness:** 3
**Presentation:** 2
**Contribution:** 3
**Rating:** 5
**Confidence:** 3

**Summary:**

This paper proposes the Recombination Flow Matching (RFM) model for protein design. RFM is inspired by evolution and combines protein segments while maintaining structural integrity. Experiments show it outperforms existing methods in generating novel proteins.

**Strengths:**

- The RFM model effectively combines principles of biological evolution, specifically recombination, into protein design. The model preserves the structural integrity of protein segments during recombination and automates the optimization of their positions.
- The experiments on benchmark datasets demonstrate the model's superiority in generating novel proteins compared to existing methods.

**Weaknesses:**

- Insufficient comparison with SOTA methods like RFDiffusion and Chroma.
- The writing needs improvement.
  - The term "movable" is undefined.
  - "T"/"R" (in Section 4.1) with different font types and sub/super-scripts represents different things, making the manuscript less readable.
  - In Section 4.1, T(s1) denotes the 1-st segment, while in Section 4.2, R(s1) denotes the timestep=1. I find it rather hard to follow.
  - In Figure 2, the training process involves predicting T(s0) given T(s1), should them be exchanged?

**Questions:**

Why do you use the average of the rotation tensor as the selected segment? Will an identity matrix work as well?

---

### Official Review · Reviewer_bMYB · 2024-11-01

**Soundness:** 3
**Presentation:** 2
**Contribution:** 2
**Rating:** 6
**Confidence:** 3

**Summary:**

Protein design holds the potential to advance drug discovery and biological research. The paper presents a flow matching framework called Recombination Flow Matching (RFM) which performs flow matching over protein residue frames to achieve recombination of natural protein segments to form novel proteins. The training was performed first on the task of unconditional protein generation and then finetuned on the task of protein segment assembly. As evaluated with scTM and novelty, the authors claim that RFM achieves SOTA performance on novel protein generation.

**Strengths:**

The idea of training a protein flow matching algorithm for protein segment assembly is interesting and can be useful to capture inter-segment relationship within proteins. This kind of work can be potentially useful for multi-motif scaffolding - generate one novel protein with multiple functional motifs.

**Weaknesses:**

1. The novelty of this algorithm is limited. The architecture of the framework is largely borrowed from previous protein flow matching papers, including protein representations and the setup of training. The addition of recombination loss is marginally novel.
2. The evaluation is not comprehensive. With scTM > 0.5 and no AF confidence reported, it is highly likely that AF models are not at all similar to the design template and yet since part of it is reminiscent of structures in training set (overlapping among AF, MPNN and this work), AF's memorization of fold nonetheless recapitulate the original pattern. It is recommend to adjust the threshold of these metrics and include AF confidence metrics.

**Questions:**

1. FrameDiff and VFNDiff are not trained for recombination. How is it a fair comparison with the proposed algorithm?
2. Is RMSD between original segment and after design segment compared?

---

### Official Review · Reviewer_YgG1 · 2024-11-03

**Soundness:** 3
**Presentation:** 3
**Contribution:** 3
**Rating:** 8
**Confidence:** 3

**Summary:**

The paper introduces the Recombination Flow Matching (RFM) model, which introduces recombinations of protein segments to the traditional flow matching model. The authors aim to develop a model that can generate more novel structures by incorporating recombination. RFM also treats proteins as rigid bodies and employs rigid body dynamics to ensure the structural integrity of the resulting proteins. The model thus can optimize for spatial arrangement of recombined segments while maintaining structural integrity. The authors conducted experiments to demonstrate the effectiveness of the model in generating novel structures, and generalizability in the protein lengths being generated.

**Strengths:**

1. Originality: the idea of applying biological recombination principles in protein design is innovative. This approach draws parallels with natural evolution, and is different from other generative model methods that emphasize on individual residues or entire protein backbones.
2. Quality: the paper is of high-quality in both the model development and evaluation. The ideas for the methodologies, e.g., using rigid-body dynamics to preserve segment integrity, are sound, well-motivated, and rigorously derived. The experiments are done on a benchmark dataset, and the results indicate a performance advantage over existing methods in novelty.
3. Clarity: the paper's writing is very clear and easy to follow. Key ideas are explained in detail. Metrics and experiments are explained well to.
4. Significance: as all the other methods in protein design, this paper has potential for great impact in that it addresses the problem of protein structure discovery, which has downstream applications in fields like drug discovery and biotechnology.

**Weaknesses:**

1. The paper focuses on generating designable and novel protein structures and does not address the direction of protein functionality at all. Would be interesting to see if this method is generalizable to designing proteins of specific functionality, which might be more relevant to downstream applications.
2. While the rigid-body dynamics employed ensure structural integrity, it might be an oversimplification of the protein evolution process. This might lead to the model not being able to produce diverse protein structures.
3. The authors compared the proposed method with several baselines. However, the comparison seems to be limited to the novelty metric. More comprehensive comparison should be performed.

**Questions:**

1. In the inference process, are the segments chosen to be combined selected manually? How is it done in the experiments and how should a downstream user select the segments?
2. What about the designabilities of the proteins generated by the baselines? How do they compare with the proposed method?
3. On line 365, RSR is written as SRS
4, The generalizability of the model is mainly discussed as the ability to generate proteins of lengths outside of the scope of the training set. Are there other aspects worth discussing?

---

### Official Review · Reviewer_2ivt · 2024-11-03

**Soundness:** 3
**Presentation:** 4
**Contribution:** 3
**Rating:** 8
**Confidence:** 1

**Summary:**

This paper proposes a novel generative model known as the Recombination Flow Matching (RFM) Model for novel protein design. RFM uses the concept of recombination from biology to create novel proteins by combing selected segments from other proteins. RFM utilizes rigid body dynamics to optimize the positions of segments within resultant protein structure. As evaluations, the authors use a dataset of protein structures to experiment with the effectiveness of RFM in comparison to prior literature. RFM outperforms baseline algorithms on both the recombination success rate as well as novelty metrics.

**Strengths:**

- Very well written paper, it was clear and a pleasure to read.
- Experiments are rigorous and provide a good set of evidence for the performance of the proposed approach.

**Weaknesses:**

- The authors don’t sufficiently detail the recombination selection algorithm, or provide an intuitive understanding (beyond an evolutionary explanation) for it being beneficial.

**Questions:**

- The selection of structures to be recombined is not made sufficiently clear. “The recombination process involves selecting segments from [P1, P2, . . . , Pm] and combining them to form P” , but this is not detailed anywhere else in the paper. (4.1 could be made a bit more detailed)
- On that note: How does the choice of structure selection algorithm (pre-recombination) affect the performance of the algorithm?
- How much can the structural novelty of proteins generated by RFM be indicative of their functional novelty? I feel as though there is a lack of discussion regarding this.
- In practice, how can we choose the number of proteins to be recombined?

---

### Comment · Area_Chair_AuuL · 2024-11-25
**Last day for reviewers to ask questions to the authors!**

Dear reviewers,

Tomorrow (Nov 26) is the last day for asking questions to the authors. With this in mind, please read the rebuttal provided by the authors and their latest comments, as well as the other reviews. If you have not already done so, please explicitly acknowledge that you have read the rebuttal and reviews, provide your updated view and score _accompanied by a motivation_, and raise any outstanding questions for the authors.

**Timeline**: As a reminder, the review timeline is as follows:
- November 26: Last day for reviewers to ask questions to authors.
- November 27: Last day for authors to respond to reviewers.
- November 28 - December 10: Reviewer and area chair discussion phase.

Thank you for your hard work,

Your AC

---

### Meta-Review · Area_Chair_AuuL · 2024-12-19

**Metareview:**

This paper received mixed reviews. Among other things, the reviewers highlight that the paper is clearly written. However, several reviewers raised concerns about insufficiently comprehensive evaluation of the proposed work. Although some reviewers have raised their scores slightly after the newly added results by the authors, there are still significant concerns remaining that require addressing. In particular, I am worried about the very limited set of metrics used to measure performance and comparison against baseline methods. During the AC-reviewer discussion phase, 5 out of 6 reviewers engaged and they all indicated to agree with this assessment, including the reviewers who had a more positively leaning score. During this discussion, reviewer aGU3 also emphasized that the use of the metrics in this paper badly interacts with the proposed method as this method has the potential to game this metric by construction. Furthermore, this reviewer also raised concerns about the novelty of this work. Finally, the 6th reviewer, who had a positive recommendation, had indicated prior to the AC-reviewer discussions to have very low confidence in their review due to a lack of context in the prior literature.
Based on all of this, I recommend to reject this paper.

**Additional Comments On Reviewer Discussion:**

See above.

---

### Decision · Program_Chairs · 2025-01-22

Reject